# Caregiver Contribution to Patient Self-Care and Associated Variables in Older Adults with Multiple Chronic Conditions Living in a Middle-Income Country: Key Findings from the ‘SODALITY-AL’ Observational Study

**DOI:** 10.3390/nursrep15100360

**Published:** 2025-10-08

**Authors:** Sajmira Adëraj, Manuela Saurini, Rocco Mazzotta, Edona Gara, Dasilva Taçi, Alta Arapi, Vicente Bernalte-Martí, Alessandro Stievano, Ercole Vellone, Gennaro Rocco, Maddalena De Maria

**Affiliations:** 1Department of Biomedicine and Prevention, University of Rome Tor Vergata, 00133 Rome, Italy; sajmira.aderaj@students.uniroma2.eu (S.A.); manuela.saurini@alumni.uniroma2.eu (M.S.); edona.gara@students.uniroma2.eu (E.G.); dasilva.taci@students.uniroma2.eu (D.T.); ercole.vellone@uniroma2.it (E.V.); 2Center of Excellence for Nursing Culture and Research, Order of Nursing Professions of Rome, 00165 Rome, Italy; alessandro.stievano@unime.it (A.S.); genna.rocco@gmail.com (G.R.); 3Albanian Order of Nurses, 1000 Tirana, Albania; alta_arapi@yahoo.com; 4Department of Nursing, Faculty of Health Sciences, University of Jaume I, 12071 Castellón, Spain; bernalte@uji.es; 5Department of Clinical and Experimental Medicine, University of Messina, 98100 Messina, Italy; 6Faculty of Nursing and Midwifery, Wroclaw Medical University, 50-996 Wrocław, Poland; 7International Center for Nursing Research Montianum Our Lady of Good Counsel, Catholic University, 1000 Tirana, Albania; 8Department of Life Science, Health, and Health Professions, Link Campus University, 00165 Rome, Italy; m.demaria@unilink.it

**Keywords:** aging, caregiver, caregiver contribution, informal caregiving, middle-income country, multiple chronic conditions, nursing, self-care

## Abstract

**Background/Objectives**: Multiple chronic conditions (MCCs) pose global health and social challenges, with caregiving often relying on family members, especially in low- and middle-income countries (LMICs). However, limited evidence exists regarding the factors influencing caregiver contribution (CC) to patient self-care among older adults with MCCs in these settings. **Aim**: The aim of this study was to examine the associations between caregivers’ and patients’ socio-demographic characteristics and patients’ clinical variables and the CC to patient self-care behaviors in adults with MCCs in an LMIC context. **Methods**: This multicenter, cross-sectional study included patient–caregiver dyads recruited from outpatient and community settings across Albania, between August 2020 and April 2021. CC was assessed using the Caregiver Contribution to Self-Care of Chronic Illness Inventory scale (CC-SCCII). Three multivariable linear regression models were used to explore associations with the three dimensions of CC to self-care maintenance, monitoring, and management. **Results**: Caregivers were mostly female, children, or spouses with a high level of education and employed. Patients were primarily female and had low education. Hypertension and diabetes were the most prevalent. Older and employed caregivers contributed less to CC to self-care maintenance, while higher education and caregiving experience increased it. Living with the patient and being a spouse reduced CC to self-care monitoring, whereas more caregiving hours and experience improved it. CC to self-care management was negatively influenced by cohabitation, presence of a second caregiver, and being a spouse, but improved with more caregiving hours. **Conclusions**: Socio-demographic and caregiving factors differently influence CC to self-care dimensions in older adults with MCCs in an LMIC. Tailored caregiver support programs are essential to enhance caregiver involvement and improve MCC patient outcomes in LMICs.

## 1. Introduction

The global shift toward an aging population presents critical challenges for healthcare systems, particularly in low- and middle-income countries (LMICs), where most older adults are projected to reside by 2050 [1]. Older adults suffer from multiple chronic conditions (MCCs), defined as the co-occurrence of at least two chronic conditions [2]. These conditions often coexist and interact, leading to complex clinical trajectories that impose a considerable burden on both patients and healthcare systems [3]. The global prevalence of MCCs is estimated at 37.2% [3], with rates in LMICs ranging from 3.2% to 90.5% [4]. MCCs have been associated with increased disability [5], impaired quality of life [6], higher rates of hospitalization and mortality [7,8], and rising healthcare utilization and costs [9,10]. In response to the increasing challenges posed by chronic diseases, the World Health Organization highlights self-care as a central approach in the management of such conditions, including multiple chronic diseases (MCCs) [11].

The Middle-Range Theory of Chronic Illness [12] defines self-care as an ongoing, active process whereby individuals sustain health, prevent worsening of disease, and address symptoms. This process consists of three interconnected components: self-care maintenance, which involves behaviors to maintain emotional and physiological balance; self-care monitoring, which is the regular observation and identification of signs and symptoms; and self-care management, which refers to the decisions and actions taken in response to these signs and symptoms. A growing body of literature on single chronic conditions has highlighted that adequate self-care can improve patient health outcomes, such as reducing rehospitalization [13], mortality [14], and quality of life [15].

However, for many older adults with MCCs, the complexity of managing multiple comorbidities—coupled with cognitive and physical limitations—makes self-care particularly challenging [16,17]. In these situations, a family caregiver (such as spouses, children or close relatives who provide unpaid care, hereafter called caregiver) can play a key role in supporting their family members in managing their chronic conditions [18], especially in LMICs due to reduced availability of healthcare resources [19]. Their contributions can directly influence treatment adherence [20], health-related quality of life [21], and patient clinical event risk [22], while also fostering self-care abilities [23]. The Situation-Specific Theory of Caregiver Contribution to Heart Failure (HF) frames these contributions into all three dimensions: self-care maintenance, self-care monitoring, and self-care management [24]. This theory conceptualized the crucial role of caregivers in supporting HF patient self-care across all three theoretical dimensions. Specifically, *CC to self-care maintenance* refers to the behaviors enacted or recommended by caregivers to help their loved one to maintain the physical and emotional stability of their chronic condition (e.g., recommend adhering to a disease-specific diet). The *CC to self-care monitoring* includes caregivers’ observational behaviors aimed at detecting new symptoms or monitoring the evolution of pre-existing symptoms (e.g., supporting or advising their loved one in monitoring the side effects of the medications they are taking). Finally, *CC to self-care management* involves the actions undertaken by caregivers to assist their loved one in managing signs and symptoms when they occur (e.g., recommending dietary modifications in response to symptom changes). Both patients’ and caregivers’ characteristics can influence self-care and CC to self-care [25].

Studies conducted in High-Income Countries (HICs) on populations of patients with single as well as chronic diseases have demonstrated that CC to self-care is influenced by different variables including characteristics of caregivers and their patients.

In HF, greater CC to self-care maintenance has been linked to being female, having higher educational level, better social support [26], cohabiting with the patient, longer duration of disease [27], and higher caregiver preparedness [28]. Higher CC to self-care monitoring was associated with female gender, having a higher education, being married and perceived higher social support [26]. Higher CC to self-care management was associated with being female and having a higher educational level, non-spousal relationship, and high caregiver preparedness [28]. Conversely, in ostomy care, CC patterns differ; greater CC to self-care maintenance relates to caregiver unemployment and fewer caregiving hours, whereas CC to self-care monitoring is associated with lower education levels and caregiver-patient cohabitation. In contrast, CC to self-care management was negatively associated with being a spousal caregiver, higher preparedness, and patient’s comorbidities [29].

In MCCs within HICs, the CC to self-care maintenance was greater with a female caregiver and in absence of a secondary caregiver [30]. Greater CC to self-care monitoring has been linked to being female, providing more hours of caregiving per week and with a lower patient education [30]. Greater CC to self-care management was associated with female caregivers and more caregiving hours per week [30].

However, despite these valuable insights from studies conducted in HICs, their applicability to LMICs remains limited. This limitation primarily stems from substantial differences in cultural norms, caregiving practices, and socioeconomic resources, which profoundly influence caregiving roles and expectations [19,31].

Managing MCCs poses additional complexities compared to single chronic disease management, creating heightened demands on caregivers, especially within LMIC contexts characterized by high MCC prevalence [4], limited healthcare infrastructure and resource scarcity [19]. In LMICs, caregivers typically assume central and often sole responsibility for disease management, significantly altering the dynamics of caregiver involvement observed in HIC settings [32]. Consequently, findings derived from HIC contexts may inadequately capture the caregiving realities in LMICs.

Given this critical gap, addressing the factors influencing CC to self-care is vital for developing culturally tailored and resource-appropriate caregiver support interventions to enhance patient health outcomes and caregiver well-being in LMICs.

This study aimed to examine the associations between caregivers’ and patients’ socio-demographic characteristics (gender, age, marital status, level of education, occupation, caregiving hours per week, years of caregiving, cohabitation, presence of a second caregiver, and caregiver–patient relationship), as well as patients’ clinical factors (number of chronic diseases), and the CC to patient self-care behaviors in adults with MCCs in an LMIC context.

## 2. Materials and Methods

### 2.1. Design

This multicenter study employed a cross-sectional approach that analyzed baseline data coming from an ongoing longitudinal study, the SODALITY study (S*elf-care* o*f patient and caregiver* d*y*a*ds in multiple chronic conditions: a long*it*udinal stud*y), examining self-care behaviors and CC in the context of MCCs. Details of the SODALITY are published elsewhere [25] and summarized below.

### 2.2. Sample and Setting

Participant recruitment was conducted through convenience sampling of patient–caregiver dyads. Between August 2020 and April 2023, participants were enrolled in outpatient and community settings in all regions of Albania. Patients and respective caregivers who presented at the clinical settings on the days established for data collection and met eligibility criteria were recruited. Caregivers were eligible if they were at least 18 years old and identified by the patient as the primary informal caregiver (e.g., a family member or close relative providing support without financial compensation). Patients were eligible if 65 years or older and diagnosed with HF, or Diabetes Mellitus (DM), or chronic obstructive pulmonary disease (COPD), coupled with one or more additional chronic conditions. Patients with cancer were excluded given the potential impact of oncological treatments (e.g., chemotherapy, radiotherapy, or surgery) on Health-Related Quality of Life [33], and patients with dementia were not considered in order to ensure reliable self-reporting. A sample estimation using G*Power vers. 3.1 software [34] confirmed that 350 dyads would adequately identify meaningful links between CC to self-care and influencing variables, assuming a medium effect size, 80% statistical power, and a significance threshold of 0.05.

### 2.3. Data Collection Process

Information was acquired by qualified nursing research assistants. Once eligible participants had been identified, the assistants described the objectives of the study and invited them to take part. Participation was confirmed after both patients and caregivers provided written informed consent. The study questionnaires were then completed independently by each member of the dyad, with support from the research assistants when visual or writing difficulties were present. To reduce potential bias, patients and caregivers were instructed to complete the instruments separately. Each data collection session required approximately 30 min.

### 2.4. Measurements

The Albanian version of the Caregiver Contribution to Self-Care of Chronic Illness Inventory (CC-SCCII-AL) [35], accessible via https://self-care-measures.com/, accessed on 27 June 2025, was used for assessing CC to self-care maintenance, self-care monitoring and self-care management. The CC-SCCII-AL is a self-administered 19-item questionnaire composed of three distinct scales: self-care maintenance (7 items, e.g., recommending the patient to do physical activity), self-care monitoring (6 items, e.g., recommending the patient to monitor the health condition), and self-care management (6 items, e.g., recommend or actually change his/her activity level when the patient has symptoms). The CC to self-care management scale can be completed only by caregivers taking care of symptomatic patients [24]. The CC-SC-CII-AL employs a five-level Likert response format, scoring from 1 (“never”) to 5 (“always”); for self-care management, the response format is employed, scoring from 1 (“not likely”) and 5 (“very likely”). Each scale produces a standardized score ranging from 0 to 100 with higher values reflecting greater CC to self-care. Scores of 70 or above are considered indicative of adequate behaviors of CC [24]. The CC-SCCII-AL was validated in an LMIC context and demonstrates strong psychometric properties, with a Comparative Fit Index of 0.98–0.99 and Root Mean Square Error of Approximation of 0.05–0.08 across its three scales, indicating excellent construct validity. Reliability is also robust, with coefficients ranging from 0.70 to 0.89 across the scales [35].

A socio-demographic questionnaire was created ad hoc by the research team to collect caregiver socio-demographic variables (i.e., age, gender, marital status, education, employment, weekly hours, years of caregiving). Patient clinical characteristics (i.e., type and number of chronic conditions) were abstracted from patient’s clinical records. The questionnaire was pilot-tested to ensure clarity and cultural appropriateness for the Albanian context. To maintain rigor in data collection, the principal investigator scheduled regular meetings with the research assistants and remained available throughout the study period.

### 2.5. Data Analysis

No missing data were identified in the study dataset. Descriptive statistics were calculated to summarize patient and caregiver characteristics, along with CC to self-care scores. Categorical variables (e.g., gender, marital status, educational attainment) were reported as frequencies and percentages, whereas continuous variables (e.g., age, weekly caregiving hours, caregiving duration, and number of chronic conditions in patients) were presented as means and standard deviations.

The identification of caregiver variables associated with CC to self-care maintenance, self-care monitoring, and self-care management was conducted in three stages. First, statistical assumptions, including normality, linearity, multicollinearity, and homoscedasticity, were assessed [36,37]. Univariate and multivariate outliers were examined using z-scores and Mahalanobis distances [38], and key assumptions were verified before conducting parametric analyses. Specifically, multicollinearity among independent variables, the normal distribution of continuous variables, and the linear relationship between each independent and dependent variable were assessed [39,40]. Second, bivariate analyses, including Chi-square (*x*^2^) tests and correlation analyses, were conducted to examine associations between caregivers’ socio-demographic, clinical patients’ variables and CC to patient self-care behaviors. Variables significantly associated with CC to self-care were identified. Third, three multivariable linear regression models were employed using maximum likelihood estimation (MLE) to analyze correlations between the caregiver variables and each CC to self-care scale and CC to patient self-care management as outcomes, adjusting for potential confounders [41]. Since the choice of variables was theoretically driven [27], all conceptually relevant predictors were simultaneously included in the models, regardless of their individual statistical significance, to ensure consistency with the study’s conceptual framework. Regression parameters were presented as unstandardized coefficients B, with 95% confidence intervals and standard errors. The coefficient of determination (R^2^) was also reported for each model. Statistical tests were two-sided, with *p*-values < 0.05 considered significant.

SPSS Statistics version 27 was used for descriptive statistics and correlation analysis. STATA version 18 was used for specific regression models, with significance set at *p* ≤ 0.05.

### 2.6. Ethical Considerations

This research was carried out in full alignment with the ethical standards set forth in the Helsinki Declaration [42]. Prior to initiation, it obtained formal clearance from the Ethics Committee of the Our Lady of Good Counsel Catholic University of Tirana (Reference Protocol: 237/2020; Approval Issued: 6 July 2020).

All individuals participating in the study provided their voluntary written consent after being fully informed about the study’s purpose, procedures, risks and benefits. Participants were also informed that they could withdraw from the study at any time during data collection; however, no participant withdrew from the study.

## 3. Results

### 3.1. Characteristics of the Sample

A total of 376 patient–caregiver dyads were included in the study, with a response rate of 100%. Socio-demographic and clinical characteristics of the participants are presented in Table 1. Caregivers had a mean age of 48.1 (SD = 15.14) years, were predominantly female (67.9%), and were mostly either the children (53.2%) or spouses (46.8%) of the patients. The majority had more than nine years of education (68.9%), were married (79.5%), and employed (61.4%). On average, caregivers had been caring for their loved ones for 10.8 years. Regarding caregiving hours per week, 38.8% of caregivers provided between 11 and 20 h per week, while 25.5% provided between 21 and 30 h per week. Patients had a mean age of 74.06 years (SD = 6.24), were predominantly female (54.25%), had less than eight years of education (65.4%), and were affected, on average, by 2.45 chronic conditions (SD = 0.66). The most frequently reported diseases were HF (88.2%) and DM (75.1%).

### 3.2. Assumptions’ Testing

Four univariate outliers were identified in the dataset. A sensitivity analysis was performed by comparing the model results with and without these outliers, revealing no substantial changes in the regression parameters. Nevertheless, they were excluded from the final analysis to enhance the skewness and kurtosis indices of the affected variables. Regression assumption testing confirmed the absence of multicollinearity among independent variables, as all Variance Inflation Factor (VIF) values were below 1.00, and correlations did not exceed the 0.80 value [43]. Linearity was deemed adequate based on a visual assessment of scatterplots. Additionally, all variables followed a normal distribution, with skewness and kurtosis values falling within the acceptable range of +1.5 to −1.5 [44].

### 3.3. Correlations Between CC to Self-Care and Caregiver and Patient Characteristics

In Table 2 the bivariate correlations are reported between CC to self-care scales and the sociodemographic and clinical variables. The bivariate correlation matrix revealed significant associations between caregiver characteristics and their contribution to patient self-care. Higher levels of CC to self-care maintenance were significantly associated with younger age (*r* = −0.170, *p* = 0.001), higher educational level (*r* = 0.134, *p* = 0.009), and unemployed status (*r* = −0.146, *p* = 0.004), suggesting that more available time and greater health literacy may enhance support in maintenance behaviors.

Regarding CC to self-care monitoring, greater contribution was positively associated with greater number of caregiving hours per week (*r* = 0.205, *p* < 0.001) and years of caregiving (*r* = 0.113, *p* = 0.029). Lower levels of CC to self-care were found among caregivers living with the patient (*r* = −0.134, *p* = 0.009) and spouse of patient (*r* = −0.181, *p* < 0.001), indicating role fatigue or redistribution of responsibilities within the household.

For the CC to self-care management, the contribution was lower among caregivers living with the patient (r = −0.116, *p* = 0.024), those sharing care with a second caregiver (r = −0.147, *p* = 0.004), and those who were the patient’s spouse (r = −0.109, *p* = 0.034), while it was positively associated with greater caregiving hours (*r* = 0.125, *p* = 0.015).

No significant associations were found between CC and patient gender, marital status, age, education level, or number of chronic conditions.

### 3.4. Variables Associated with CC to Self-Care Maintenance, Monitoring and Management

Model 1, examining variables associated with CC to self-care maintenance, accounted for 49.7% of the variance (R^2^ = 0.497, F = 5.773, *p* < 0.001) (Table 3). Caregiver occupation (β = −0.487, *p* = 0.022), years of caregiving (β = 0.111, *p* = 0.048) and caregiver age (β = −0.058; *p* = 0.018) were significantly associated with CC to self-care maintenance. Specifically, being unemployed, younger in age, and having more years of caregiving were linked to higher levels of CC to self-care maintenance.

Model 2, assessing factors associated with CC to self-care monitoring, explained 50.5% of the variance (R^2^ = 0.505, F = 5.032, *p* < 0.001). A greater number of caregiving hours per week was positively associated with CC to self-care monitoring (β = 0.071, *p* = 0.001) while being the patient’s spouse was negatively associated (β = −0.251, *p* = 0.003).

Model 3, focused on CC to self-care management, accounted for the largest proportion of explained variance (R^2^ = 0.681, F = 6.287, *p* < 0.001). Higher caregiving hours were positively associated with CC to self-care management (β = 0.041, *p* = 0.036), while the presence of a second caregiver (β = −1.155, *p* = 0.005) and being a spouse (β = −0.159, *p* = 0.036) were negatively associated.

In all three models, none of the patient-related variables were significantly associated with CC to self-care dimensions.

## 4. Discussion

This study aimed to examine the variables associated with CC to self-care among older adults with multiple MCCs in an LMIC. To the best of our knowledge, this is the first study that explored the association of the socio-demographic characteristics of MCCs patient and caregiver living in a LMIC. Our findings confirm that the socio-demographic characteristics of MCC patients and their caregiver can influence the CC to self-care behaviors in LMICs. Our findings indicate that socio-demographic characteristics of patient and caregiver impact the CC to self-care behaviors differently. These results provide new insights for LMICs and complement existing evidence from HICs, highlighting the need for context-specific strategies to support informal caregivers.

An important feature of our sample was the predominance of female caregivers (67.9%), which reflects Albania’s family-oriented caregiving norms, where women often assume primary caregiving roles due to traditional gender expectations [45]. This pattern is consistent with cultural practices observed in many LMICs, where caregiving remains predominantly a female responsibility shaped by social norms and family structures. Recognizing this cultural dimension is essential for interpreting our findings and for designing interventions that are both gender-sensitive and context-appropriate.

### 4.1. Variables Associated with CC to Self-Care Maintenance

This study found that employment status, caregiver age, and the number of caregiving hours were significantly associated with CC to self-care maintenance. Specifically, unemployed caregivers contributed more extensively to self-care maintenance tasks, likely due to their greater availability of time to engage in routine activities essential for maintaining patients’ health. This finding aligns with evidence from HICs [29], reinforcing the importance of available time as a pivotal resource for effective caregiver support, particularly in LMICs. Younger caregivers were also found to have greater CC to self-care maintenance, possibly due to fewer physical limitations and higher emotional resilience compared to older caregivers, who might face barriers in consistently managing daily caregiving tasks. This finding is consistent with existing research from both high- and low-income settings [30,45]. Furthermore, younger caregivers may benefit from increased physical capabilities, access to health-related information, and more familiarity with contemporary caregiving practices, further enabling them to actively support self-care behaviors. Additionally, a greater year of caregiving hours was positively associated with higher CC to self-care maintenance. Spending more hours on caregiving roles likely enhances caregivers’ practical skills, emotional adaptability, and overall confidence in performing routine self-care behaviors. This relationship supports the theoretical framework of self-care and CC to self-care proposed by Riegel and Vellone [12,24], highlighting caregiving experience and dedicated time as key components that bolster effective support in self-care behaviors. Especially in LMICs, where formal caregiver training opportunities are limited, experiential learning gained through extensive caregiving hours represents a crucial resource, facilitating sustained and competent support for patient self-care maintenance.

### 4.2. Variables Associated with CC to Self-Care Monitoring

Greater contribution to symptom monitoring was associated with higher weekly caregiving hours and being a non-spousal caregiver. These findings are consistent with evidence from HICs showing that more time spent caregiving enhances caregiver vigilance and symptom awareness [30].

In LMICs, extended caregiving hours often indicate gaps in formal care services and contribute to increased caregiver burden [46], highlighting the need for interventions that enhance monitoring skills while promoting sustainable caregiving. Spousal caregivers were less likely to contribute to monitoring, possibly due to role fatigue, age-related limitations, personal health issues, or low self-esteem that may hinder the objective recognition of symptom changes [47]. In contrast, non-spousal caregivers, such as sons or daughters, may exhibit higher contributions due to their potentially stronger physical and cognitive capacities, as well as different relational dynamics that allow for more objective symptom observation.

In LMICs, where caregiving is typically informal and lacks systemic support, aging spouses often struggle to meet the complex demands of MCCs, particularly in multigenerational households with high expectations and limited formal resources. These differences in family roles, combined with gender patterns, suggest that educational approaches should account not only for the caregiver’s sex (e.g., male patient–female caregiver, female patient–female caregiver, etc.) but also for the specific patient–caregiver relationship (e.g., mother patient–child caregiver). Tailoring interventions to these dyadic configurations may enhance their effectiveness.

### 4.3. Variables Associated with CC to Self-Care Management

CC to self-care management was positively associated with higher weekly caregiving hours, being non-spousal and the absence of a second caregiver. Higher caregiving intensity, reflected in more hours per week, was associated with greater CC to self-care management. This aligns with findings from HICs [30], where more time spent with the patient enhances symptom recognition and timely response. In LMICs, where formal care is limited, caregiving intensity often substitutes clinical oversight, positioning informal caregivers as primary responders to symptom exacerbations. While greater involvement may reflect higher engagement, it also raises the risk of caregiver burnout, underscoring the need for targeted education and community-based interventions. The lower contribution observed among spouses aligns with previous studies and may reflect the dual burden of managing their own health alongside their caregiving role [29]. Conversely, sons or daughters as caregivers may demonstrate greater CC due to higher health literacy, fewer personal health constraints, and a filial duty that motivates proactive management [48].

These findings highlight the vulnerability of spousal caregivers in LMICs, particularly in multigenerational households with limited external support. The presence of a second caregiver may lead to fragmented responsibilities and decreased accountability, especially in informal caregiving settings lacking coordination, unlike more structured systems in HICs. Additionally, children or other non-spousal caregivers may be more proactive in responding to symptom exacerbations, due to better health literacy or greater physical and cognitive capacities [49]. In the context of LMICs, where institutional support is minimal, the role of the sole caregiver remains critical for effective symptom management. However, this can also lead to chronic overload, highlighting the need for policies that support this group through education, social assistance, and regular respite opportunities.

In line with the theoretical model of self-care in chronic illness developed by Riegel [12] and subsequently on CC to self-care by Vellone [24], our findings suggest that interventions should focus on educating and empowering caregivers in all three dimensions: maintenance, monitoring, and management. Variables such as caregiving experience, hours of care, the caregiver–patient relationship, and cohabitation influence each self-care dimension in different ways. While in HICs the impact of these factors is often mitigated by structured healthcare systems, in LMICs, their effect is more direct and pronounced due to the lack of formal caregiving infrastructure.

While socio-demographic variables (e.g., caregiver age, employment, caregiving hours, cohabitation, presence of a second caregiver, and caregiver–patient relationship) showed statistically significant correlations with CC to self-care (r = 0.10–0.20, Table 2) and modest regression model explanatory power (R^2^ = 0.497–0.681, Table 3), their practical and clinical impact in LMICs settings appears limited due to the weak magnitude of these associations. These findings should therefore be interpreted with caution and underscore the need for further research to identify variables capable of better explaining CC in resource-constrained contexts. Targeted interventions remain essential to strengthen the caregiver’s role and improve patient outcomes, but their design should take into account the modest explanatory power of socio-demographic factors.

Finally, we did not find any association between CC to self-care dimensions and patient characteristics such as gender, education, number of diseases, and disease type. Presumably, there are other specific variables missing that can influence CC to MCCs self-care. Therefore, further research is needed to deepen the understanding of CC to MCCs self-care and its associated variables in LMICs.

### 4.4. Strengths, Limitations, and Future Research

A key strength of this study is the inclusion of a large sample, along with the use of a validated and culturally adapted instrument (CC-SCCII-AL), ensuring methodological rigor and reliability in assessing CC to self-care [35]. This rigor is especially important given the complex and culturally embedded nature of caregiving in these contexts. However, several limitations should be acknowledged. The cross-sectional design limits causal inferences and the ability to track changes over time.

Additionally, the sample’s predominance of female caregivers (67.9%) reflects Albania’s family-oriented caregiving norms, where women often assume primary caregiving roles due to traditional gender expectations [46]. This pattern aligns with cultural practices in many LMICs, where caregiving is frequently a female-dominated responsibility, shaped by societal norms and family structures. Recognizing this cultural dimension is essential for interpreting our findings and for designing interventions that are both gender-sensitive and context-appropriate.

## 5. Conclusions

This study shows that socio-demographic and caregiving-related factors variably affect CC to self-care maintenance, monitoring, and management in older adults with MCCs in a LMIC. These findings emphasize the need for targeted support for older, employed, and spousal caregivers, alongside better coordination within caregiving networks to enhance self-care in LMICs settings.

## Figures and Tables

**Table 1 nursrep-15-00360-t001:** Socio-demographic and clinical characteristics of the participants (N = 376 MCC patients and their caregivers).

Variable	Caregiver	Patient	*p*-Value
	M ± SD	M ± SD	
Age (years)	48.10 (15.14)	74.06 (6.24)	<0.001
	N (%)	N (%)	
Gender			
Female	255 (67.90)	204 (54.25)	0.004
Male	121 (32.10)	172 (45.74)	
Marital status			
Married	299 (79.50)	255 (67.81)	0.007
Single/Divorced/Widowed	77 (100)	121 (32.18)	
Education level (in years)			
<8	117(31.10)	246 (65.42)	<0.001
≥9	259 (68.90)	130 (34.57)	
Employment status			
Employed	231 (61.44)	11 (2.92)	0.003
Retired/Unemployed	145 (38.56)	365 (97.07)	
Perceived income			
Less than needed	46 (12.30)	75 (19.94)	<0.001
More than needed	330 (87.70)	15 (3.98)	
Relationship patient-caregiver			
Son/daughter		200 (53.19)	
Spouse/partner		176 (46.81)	
Living with patient			
Yes		142 (37.77)	
No		234 (62.23)	
Caregiving hours per week			
0–10	68 (18.08)		
11–20	146 (38.82)		
21–30	96 (25.53)		
>30	66 (17.55)		
Years of caregiving			
0–10	323 (85.90)		
11–20	46 (12.23)		
21–30	5 (1.32)		
>30	2 (0.53)		
Presence of a second caregiver			
Yes	14 (3.72)		
No	362 (96.27)		
		M (SD)	
Number of Chronic Conditions		2.45 (±0.66)	
		N (%)	
Type of chronic disease			
HF		337 (88.2)	
DM		287 (75.1)	
COPD		52 (13.61)	
Other chronic diseases		89 (23.6)	

Legend: COPD, Chronic Obstructive Pulmonary Disease; DM, Diabetes Mellitus; HF, Heart Failure; M, Mean; SD, Standard Deviation. Note. The *p*-values indicate statistical comparisons between the groups.

**Table 2 nursrep-15-00360-t002:** Results of bivariate correlations between CC to patient self-care and participants characteristics (N = 376 MCC patients and their caregivers).

Variable	CC to Patient Self-Care Maintenance r (*p*-Value)	CC to Patient Self-Care Monitoring r (*p*-Value)	CC to Patient Self-Care Management r (*p*-Value)
**Caregiver variables**			
Gender (female = 1, male = 0)	0.015 (0.769)	0.054 (0.299)	0.031 (0.546)
Age (years)	**−0.170 (0.001)**	−0.043 (0.406)	−0.063 (0.224)
Marital status (married = 1, single/divorced/widowed = 0)	−0.020 (0.706)	0.030 (0.563)	0.024 (0.649)
Level of education in years (≥9 = 1, l < 8 = 0)	**0.134 (0.009)**	0.047(0.362)	0.038 (0.459)
Occupation (employed = 1, unemployed = 0)	**−0.146 (0.004)**	0.020 (0.701)	−0.050 (0.336)
Caregiving hours per week	0.09 (0.078)	**0.205 (<0.001)**	**0.125 (0.015)**
Years of Caregiving	0.076 (0.142)	**0.113 (0.029)**	0.056 (0.277)
Living with the patient (yes = 1, no = 0)	−0.075 (0.148)	**−0.134 (0.009)**	**−0.116 (0.024)**
Presence of second caregiver (yes = 1, no = 0)	−0.084 (0.105)	−0.059 (0.257)	**−0.147 (0.004)**
**Patient variables**			
Gender (female = 1, male = 0)	0.049 (0.339)	−0.049 (0.339)	−0.037 (0.479)
Age (years)	−0.016 (0.751)	−0.017 (0.746)	0.005 (0.917)
Level of education in years (≥9 = 1, <8 = 0)	−0.012 (0.822)	0.045 (0.388)	−0.010 (0.843)
Relationship between patient and caregiver (spouse = 1, child = 0)	−0.036 (0.481)	**−0.181 (<0.001)**	**−0.109 (0.034)**
Number of chronic diseases	−0.039 (0.455)	−0.064 (0.215)	−0.079 (0.125)

Note. The Caregiver Contribution (CC) to self-care inventory was used to assess the CC to self-care maintenance, CC to self-care monitoring and CC to self-care management.

**Table 3 nursrep-15-00360-t003:** Results of multivariable linear regression models showing the association between caregiver and patient characteristics and CC to self-care maintenance, monitoring and management (N = 376 caregivers).

Independent Variables	Model 1			Model 2			Model 3		
	CC to Self-Care Maintenance			CC to Self-Care Monitoring			CC to Self-Care Management		
	β	b	*p* Value	β	b	*p* Value	β	b	*p* Value
**Caregiver variables**									
Gender (female = 1, male = 0)	0.396	0.640	0.537	0.165	0.560	0.768	0.143	0.499	0.774
Age (years)	−0.058	0.024	**0.018**	−0.038	0.021	0.079	−0.025	0.019	0.191
Marital status (married = 1, single/divorced/widowed = 0)	0.467	0.589	0.429	0.652	0.515	0.207	0.502	0.458	0.274
Level of education (high = 1, low = 0)	0.462	0.257	0.073	0.303	0.225	0.179	0.161	0.200	0.422
Occupation (employed = 1, unemployed = 0)	−0.487	0.212	**0.022**	0.009	0.185	0.961	−0.193	0.165	0.242
Caregiving hours per week	0.048	0.025	0.058	0.071	0.022	**0.001**	0.041	0.020	**0.036**
Years of caregiving	0.111	0.056	**0.048**	0.056	0.049	0.254	0.026	0.043	0.556
Living with the patient (yes = 1, no = 0)	−0.739	0.529	0.163	−0.819	0.463	0.078	−0.699	0.412	0.091
Presence of second caregiver (yes = 1, no = 0)	−0.667	0.525	0.205	−0.536	0.460	0.245	−1.155	0.409	**0.005**
**Patient variables**									
Gender (female = 1, male = 0)	0.523	0.626	0.404	0.020	0.548	0.971	−0.084	0.487	0.864
Age (years)	−0.033	0.048	0.500	−0.021	0.042	0.623	−0.001	0.038	0.979
Level of education (high = 1, low = 0)	−0.245	0.250	0.326	−0.043	0.218	0.843	−0.152	0.194	0.435
Relationship between patient and caregiver (spouse = 1, child = 0)	−0.124	0.097	0.202	−0.251	0.085	**0.003**	−0.159	0.076	**0.036**
Number of chronic diseases of patients	−0.361	0.438	0.410	−0.490	0.383	0.201	−0.535	0.341	0.117
**Constant**	38.899	4.234	<0.001	25.665	3.706	<0.0001	27.905	3.298	<0.001
**Mean dependent variable**	34.298			22.809			24.024		
**R-squared**	0.497			0.505			0.681		
**F-test**	5.773			5.032			6.287		

Abbreviations. β, standardized coefficient; b, unstandardized coefficient.

## Data Availability

Dataset is available upon request from the authors.

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
