# Peer review of "Caregiver Contribution to Patient Self-Care and Associated Variables in Older Adults with Multiple Chronic Conditions Living in a Middle-Income Country: Key Findings from the ‘SODALITY-AL’ Observational Study"

_nursrep, 2025, doi:10.3390/nursrep15100360_

Round 1
Reviewer 1 Report
Comments and Suggestions for Authors
Nursing reports - Caregiver Contribution to Patient Self-Care and Associated Variables in Older Adults with Multiple Chronic Conditions living in a Middle-Income Country
Dear Authors,
Thank you for the opportunity to review your manuscript. I appreciate the effort and dedication that went into your work. Below are detailed comments and suggestions intended to support the improvement of your manuscript, aligned with the STROBE guidelines and best practices for cross-sectional research.

Reviewer 2 Report
Comments and Suggestions for Authors
Dear Authors,
Thank you for submitting your manuscript. Please refer to the attached file for details. I hope that my comments will contribute, even in a small way, to the improvement of your manuscript.
Best regards,

Reviewer 3 Report
Comments and Suggestions for Authors
Thank you for this manuscript, which I enjoyed reading. I mainly have some general comments, since the analytical method is not my expertise.
General comments:
Introduction and throughout: while caregiver is the accepted phrase to describe family member providing care, we should consider their family roles as well, particularly given the impact of care giving on their relationship with the person they support. E.g. in line 148 you refer to 'informal caregivers and their patients' are they not a spouse or parent rather than their patient? This choice of language has implications to how we view their relationship (and maybe this is intentional ) from a family relationship to a clinical relationship. However, if this is the case, this should be further explored in the discussion, or rephrased.
I cannot comment on the statistical analysis as this is outside of my expertise.
Table 1: Title missing
Discussion/ conclusion: In line with my comment above - consideration of the care givers family relationship to the person they are caring for is relevant - might a son/ daughter have a different CC to a parent than a spouse? We know that gender clearly plays a role in caregiving, as also confirmed in your study - might this also be relevant for different educational approaches and how we support the caregiver?
Line 53: Please replace the term 'elderly' as this is associated with negative stereotypes. Use 'older person' or 'older adult'.
